# Circulating Endothelial Cell Levels Correlate with Treatment Outcomes of Splanchnic Vein Thrombosis in Patients with Chronic Myeloproliferative Neoplasms

**DOI:** 10.3390/jpm12030364

**Published:** 2022-02-27

**Authors:** Giulio Giordano, Mariasanta Napolitano, Michele Cellurale, Paola Di Carlo, Gerardo Musuraca, Giorgia Micucci, Alessandro Lucchesi

**Affiliations:** 1Internal Medicine Division, Hematology Service, Regional Hospital “A. Cardarelli”, 86100 Campobasso, Italy; giuliogiordano@hotmail.com (G.G.); michele.cellurale@asrem.org (M.C.); 2Department of Health Promotion, Mother and Child Care, Internal Medicine and Medical Specialties and Haematology Unit, University Hospital “P. Giaccone”, 90127 Palermo, Italy; mariasanta.napolitano@unipa.it; 3Department of Health Promotion, Mother, and Child Care, Internal Medicine and Medical Specialties and Infectious Disease Unit, University Hospital “P. Giaccone”, 90127 Palermo, Italy; paola.dicarlo@unipa.it; 4Hematology Unit, IRCCS Istituto Romagnolo per lo Studio dei Tumori (IRST) “Dino Amadori”, 47014 Meldola, Italy; gerardo.musuraca@irst.emr.it

**Keywords:** myeloproliferative neoplasms, portal vein thrombosis, circulating endothelial cells, recanalization

## Abstract

Circulating endothelial cells (CECs) are viable, apoptotic or necrotic cells, identified by CD 146 surface antigen expression, considered a biomarker of thrombotic risk, given their active role in inflammatory, procoagulant and immune processes of the vascular compartment. Growing evidence establishes that CECs are also involved in the pathogenesis of several hematological and solid malignancies. The primary aim of this study was to verify if CEC levels could predict both the course and treatment responses of splanchnic vein thrombosis (SVT), either in patients affected by myeloproliferative neoplasms (MPNs) or liver disease. Thus, a retrospective multicenter study was performed; fifteen patients receiving anticoagulant oral treatment with vitamin k antagonists (VKA) for SVT were evaluated. Nine patients were affected by MPN, and all of them received cytoreduction in addition to anticoagulant therapy; four of these patients had primary myelofibrosis (PMF) and were treated with ruxolitinib (RUX), and one patient with primary myelofibrosis, two patients with essential thrombocythemia (ET), and two patients with polycythemia vera (PV) were treated with hydroxyurea (HU). Six patients affected by liver diseases (three with liver cirrhosis and three with hepatocellular carcinoma) were included as the control group. CECs were assayed by flow cytometry on peripheral blood at specific time points, for up to six months after enrollment. The CEC levels were related to C-reactive protein (CRP) levels, splenic volume reduction, and thrombus recanalization, mainly in MPN patients. In patients with liver cirrhosis (LC) and hepatocellular carcinoma (HCC), for which the mechanism of SVT development is quite different, the relationship between CEC and SV reduction was absent. In conclusion, the CEC levels showed a significant correlation with the extent of venous thrombosis and endothelial cell damage in myeloproliferative neoplasm patients with splanchnic vein thrombosis. Although preliminary, these results show how monitoring CEC levels during cytoreductive and anticoagulant treatments may be useful to improve SVT outcome in MPN patients.

## 1. Introduction

Circulating endothelial cells (CECs) are a specific population of endothelial cells, identified by CD146 surface antigen expression, using flow cytometry; CECs are usually present at a very low percentage in the peripheral blood (0.01% of the mononuclear cells) [1]. CECs are not only involved in angiogenesis, but they actively take part in inflammatory, procoagulant and immune processes of the vascular compartment, in equilibrium with vessel endothelial cells [2]. Given their role in inducing the expression of proinflammatory signals by endothelial cells, CECs may be considered as biomarkers of vascular injury and thrombotic risk [3].

Growing evidence suggests that CECs play an important role in the pathogenesis of several hematological diseases; high levels of mature, resting, activated circulating endothelial cells and CECs have been observed within the course of both solid and hematological neoplasms [4,5,6,7,8,9].

With special regard to myeloproliferative neoplasms (MPNs), an increased CEC count may reveal endothelial damage in patients with VTE, and express different endothelial activation phenotypes, depending on the disease state. In patients with MPN, after an episode of deep vein thrombosis (DVT), the CEC count rapidly increases, when compared with healthy individuals. In a subgroup of subjects with acute DVT, the CEC levels progressively decreased within 9–15 months after the acute event [10].

Chronic Philadelphia-negative myeloproliferative neoplasms (MPNs) are hematological malignancies that include the following three main entities: polycythemia vera (PV), essential thrombocythemia (ET), and primary myelofibrosis (PMF). These disorders share common features, especially the high risk of arterial and venous thrombosis that can be severe, recurrent, and life threatening. Venous thrombosis in patients with MPN may occur at unusual sites, such as in the splanchnic (SVT) or cerebral (CVT) veins [11].

SVTs in MPNs show a prevalence of 5–10% in PV patients and 9–13% in ET patients. SVTs occur less frequently in patients with PMF, with a prevalence of 0.6–1% [12,13].

SVT may involve the portal vein, as in portal vein thrombosis (PVT), the hepatic venous system, as in Budd–Chiari syndrome (BCS), the splenic vein, as in splenic vein thrombosis (SpVT), or the mesenteric vein, as in mesenteric vein thrombosis (MVT) [14]. PVT is the most common type of SVT (40%), while BCS occurs less frequently (5%) [11]. Among 604 patients listed in the International Registry of SVT, the most common local risk factors for SVT were liver cirrhosis (27.8%) and solid cancer (22.7%), while MPNs were the most frequent systemic cause of SVT (8.2%) [15].

Occlusion of the splanchnic vessels, with consequential portal hypertension, together with the extramedullary hematopoiesis in MPNs, contribute to a progressive splenomegaly.

More than 95% of patients with PV [16], and around 50% of patients with PMF and ET [17], carry the JAK2V617F gene mutation, which confers an increased risk of arterial and venous thrombosis in patients with MPN [18,19,20,21], and, importantly, it has been confirmed as an independent predictive risk factor for SVT [22].

Disease burden modulation appears to be a winning strategy for preventing vascular manifestations in patients with MPN. Hydroxyurea (HU) is a cytoreductive agent that inhibits ribonucleotide reductase, and is administered to MPN patients for cell count control.

HU probably owes its antithrombotic effects to its ability to inhibit p-selectin-induced tissue factor (TF) expression on polymorphonuclear cells [23,24].

The potential antithrombotic effects of HU have also been studied for the treatment of sickle cell disease [25,26].

The pathogenetic model of the “circulating wound”, recently proposed by our research team, would, thus, undergo dampening [27]. A similar benefit seems to be offered by anti-JAK1/2 targeted therapies (e.g., ruxolitinib) that are capable of calming the cytokine storm, modulating the expression of endothelial integrins, and reducing the pro-adhesive capacities of circulating cells [28].

It is proven that ruxolitinib is effective in reducing splenomegaly and controlling MPN-related symptoms compared with a placebo (in MF, COMFORT-I study) or with the best available therapy (in both MF and PV, COMFORT-II and RESPONSE studies, respectively) [18,19,28]. However, the impact of ruxolitinib on the course of SVT in patients with MF is still unsatisfactorily explored.

The primary aim of the current study was to assess if CEC levels were influenced by ruxolitinib (RUX) treatment in patients with MF at disease onset, or by hydroxyurea (HU) therapy in patients with MPN and SVT. The secondary objective of this study was to evaluate the correlation between CEC levels and the following parameters: D-dimer (DD), C-reactive protein (CRP), rate of thrombosis recanalization, splenic volume (SV), and improvement in systemic symptoms. We also explored the influence of treatments on ameliorating the venous congestion in the splanchnic district, thus determining stabilization/improvement of pre-existing esophageal varices or preventing the formation of new esophageal varices.

## 2. Patients and Methods

In this dicentric retrospective study, arising from a collaboration between the Division of Hematology and the Division of Gastroenterology (Campobasso, Italy), we enrolled a total of 15 adult patients with SVT, of which 9 were affected by MPN (PV, ET or MF, diagnosed according to 2008 WHO criteria) [21], while 6 patients with hepatic disease (liver cirrhosis-LC or hepatocarcinoma-HC) were analyzed as the control group. SVT may have preceded or followed the diagnosis of MPN, LC or HC. Data were collected from January 2014 to January 2020.

The study was conducted in agreement with the Declaration of Helsinki; the samples were collected and processed after obtaining informed consent from patients, and the data collection was approved by the local institutional review board (IRB). The distribution between male and female patients was 1:2. The median age was 45 years (range 35–65). Patients were affected by the following MPNs: MF (N = 5, 33.3%), PV (N = 2, 13.3%), ET (N = 2, 13.3%); the hepatic diseases were LC (N = 3, 20%) and HC (N = 3, 20%). All patients with PV, ET and MF carried the JAK2 V617F gene mutation. Only 2 patients (13.3%) presented with the factor V Leiden gene mutation, at heterozygous state. At SVT onset, median hemoglobin (Hb) levels were 11.5 g/dL (range 9–14.5), median white blood cell (WBC) count was 9000/mmc (range 3000–15,000), median platelet (PLT) count was 90,000/mmc (range 30,000–250,000), median D-dimer (DD) level was 5200 µg/mL (FEU) (range 3000–12,000), median C-reactive protein (CRP) concentration was 80 mg/L (range 20–200), and median circulating endothelial cell (CEC) levels were 1800/µL (range 1000–3000). Each reported parameter was assayed at the following time points: study entry, 1 month, 3 months and 6 months at follow-up visits. All patients had PVT, which was associated with splenic vein thrombosis (SpVT) and mesenteric vein thrombosis (MVT), respectively, in 8/15 (53%) and 1/15 (6%) of the cases. Systemic symptoms, including weight loss, fever, night sweats and fatigue, were reported by 14/15 (93%) of the patients. SVT was diagnosed within a median of 3 months (range 0–24) before ruxolitinib (RUX) or hydroxyurea (HU) treatment. The median length of the spleen below the costal margin was 15 cm, ranging from 10 to 25 cm; all patients had palpable hepatomegaly.

We divided patients into the following three groups: group A included patients with PMF and SVT at disease onset or after diagnosis, who were treated with vitamin k antagonists (VKA) and RUX as cytoreductive therapy, which was allowed as the treatment in these patients, as they all had an intermediate-1 IPSS risk score [29]; group B included patients with MPN (PV, ET and MF) and SVT that occurred at disease onset or after diagnosis, who were treated with VKA and HU—the median duration of anticoagulant therapy for group A and B was 3 months (range 65–120 days); group C included patients with LC or HCC and SVT that occurred at disease onset or after diagnosis, who were treated with vitamin K antagonists (VKA) only.

Patients’ characteristics from the three groups are shown in Table 1. During the disease course, patients from group A received a median RUX dose of 40 mg/die (range 15–40), and group B received a median HU dose of 1000 mg/die (range 500–2000). The dosage of the drug, in accordance with the technical data sheet, was adjusted, taking into account the hematological values and the patient’s weight.

Parameters and data of interest were collected for six months after SVT onset. At the 6-year follow-up, three patients with HCC had died; the remaining were still alive and undergoing treatment.

CECs were detected using flow cytometry, and were defined as negative for CD45 and CD133 cells and positive for CD146 cells [10]. Two hundred microliters of peripheral blood in sodium heparin was labeled with 10 mol of a panel of fluorescein isothiocyanate (FITC) R-phycoerythrin (R-PE), or the peridinin chlorophyll protein-conjugated antibodies anti-CD45, anti-CD133 and anti-CD146, for 20 min at room temperature. After conjugation, red blood cells were lysed by incubating in FACS lysing solution (Becton Dickinson, San Jose, CA, USA) for 15 min at room temperature. White blood cell pellets were then washed twice in FACS Flow solution (Becton Dickinson). Appropriate analysis gates were adopted to enumerate total CEC.

Five-parameter, three-color flow cytometry was performed with a FACScan flow cytometer with a 15 mW argon laser (excitation at 488 mm) (Becton Dickinson). Cells stained with isotypic controls for IgG1–FITC or R-PE were used as negative controls. At least 100,000 (typically 300,000) cells per sample were acquired; analyses were considered informative when adequate numbers of events (100) were collected in the CEC enumeration gates [4]. Data were analyzed with CellQuest software (Becton Dickinson).

Splenomegaly reduction was assessed using ultrasound (US), to determine the percentage reduction in splenic dimension below the lower costal margin, from the initial dimension.

Thrombus recanalization was assessed as the percentage reduction in thrombus extension from the initial dimension, measured using US-Doppler. Data were collected at SVT diagnosis and 1, 3, and 6 months after SVT onset.

In order to evaluate statistical correlation between the variables considered, Pearson’s linear correlation test was adopted, whereas to analyze differences among the three groups of treatment, an ANOVA test was performed. Differences between groups A and B were evaluated by *t*-test.

## 3. Results

The values of the variables considered in each treatment group during the observation period are summarized in Table 2. The CEC levels are directly related to CRP in all three treatment groups (r = 0.7 in group A, r = 0.9 in group B, and r = 0.9 in group C). The CEC levels showed a direct correlation with D-dimer in patients treated with RUX (r = 0.8) and HU (r = 0.98), but not in patients affected by LC or HCC (r = 0.3) (Figure 1, Figure 2 and Figure 3).

On the other hand, the CEC levels are inversely related to the degree of thrombosis recanalization in all the treatment groups (r = −0.7 in the RUX group, r = −0.8 in the HU group, and r = −0.9 in the VKA group), and to the splenic volume reduction (r = −0.96 in the RUX group and r = −0.9 in the HU group), except for patients with LC and HCC, in which the spleen volume is not reduced, even after VKA therapy (Table 3).

Splenic volume reduction is directly related to the size of thrombosis recanalization in the RUX group (r = 0.8) and the HU group (r = 0.94). Of note is that CRP is inversely related to the splenic volume reduction in the RUX group (r = −0.99) and in the HU group (r = −0.67).

When we evaluated the differences between the three treatment groups (RUX vs. HU vs. VKA) with ANOVA testing, we only found significant differences in the decreases in CEC levels (*p* = 0.04) and splenic volume (*p* = 0.03). Thrombus recanalization started after one month of VKA and RUX or HU administration, and produced higher values in the RUX group, reaching a plateau after 3 months for the RUX group and after 6 months for the HU group. The group treated with RUX showed greater patency of the affected vessels than the HU group (*p* = 0.01). The highest value of thrombosis recanalization, expressed as a percentage (%), and SV reduction was observed in the RUX treatment group.

## 4. Discussion

CECs are viable, CD146-positive, and CD45- and CD133-negative cells that may mirror endothelial cell injury; they are frequently increased during the course of several inflammatory conditions [2], where they have also been directly related to CRP levels. Even in patients with acute myocardial infarction, a condition in which the inflammatory component plays an important role, a positive correlation between CECs and CRP was found, independent of age, gender, serum cholesterol levels, hypertension, obesity, history of cardiovascular disease, or smoking [30].

The high median levels of CRP observed in our cohort, given the young median age of the enrolled patients, are probably not due to aging-related diseases (i.e., cardiovascular disease, hypertension, diabetes mellitus, and kidney disease), but to inflammation related to the neoplasm. Of interest is the speculation of whether MPNs are interpretable as “vascular disease” because they share pathogenetic traits that are typical of pro-thrombotic conditions, such as atherosclerosis, cancer, and hereditary thrombophilia [31].

This is not surprising, since female sex and young age, usually around 40 years, are frequent characteristics of patients affected by MPN with SVT [22]; this suggests that in MPN, the risk factors for SVT are different from those historically considered for classical VTE (e.g., age >60 years and a history of thrombosis). The pathogenetic mechanism of SVT is probably different, and has to be investigated by the unique interaction between activated JAK2-positive cells and the peculiar splanchnic district, a low-flow vascular bed that can, itself, harbor the JAK2 mutation [22]. The JAK2 mutation plays an “all or nothing” pro-thrombotic effect, also with low allelic burden, and can be found in SVT as an isolated positivity; the so-called “solitary JAK2” is more frequently associated with PVT, and up to 52% of the patients could manifest overt MPN during follow-up [22]. Further specific gender risk factors for thrombosis may be involved. The inflammatory profile of women may be influenced by estrogens. Some of the proposed mechanisms of how estrogens regulate inflammation include responses mediated via nuclear-/membrane-bound estrogen receptors, stimulation of Toll-like receptor ligands on dendritic cells and macrophages, resulting in increased expression of proinflammatory cytokines, decreasing inhibitory PI3K signaling and Akt phosphorylation in macrophages, and effects on NF-κB transcriptional complexes [32].

An estimated 40% of BCS patients and 28% of PVT patients without an MPN diagnosis carry the JAK2 V617F gene mutation [11], while 71–100% of patients with overt MPN and SVT test positive for JAK2 V617F, which is significantly higher than the JAK2 positivity reported for other VTEs within the course of MPN [33]. Our study confirmed this trend (JAK2 V617F mutation detected in all the enrolled subjects), and showed a direct correlation between CEC and CRP levels.

The JAK2 V617F mutation induces an inflammatory phenotype mediated by cytokine, chemokine and growth-hormone signaling, thus contributing to inflammation, myeloproliferation, thrombosis development, splenomegaly and constitutional symptom onset in patients with MPN [31,34].

Moreover, mutant endothelial cells might contribute to the pathogenesis of SVT. In fact, liver endothelial cells carrying the JAK2 mutation were found in two BCS patients [35], and the JAK2 mutation was also detected in spleen endothelial cells and circulating endothelial progenitor cells [36]. Mutated CECs displayed increased adherence to normal mononuclear cells and abnormal activation of the JAK/STAT pathway [37].

Taking into account that the portal venous system is exposed to gut-derived antigens and inflammatory stimuli that may influence its immunogenicity, we can understand how vessels are more vulnerable to activated platelets and to high blood viscosity. We can, therefore, also understand why the JAK2 V617F mutation, affecting not only cell numbers, but also characteristics, may lead to SVT [38]. All these data support the hypothesis that CECs are probably not only an indicator of systemic inflammation, but also a cause of SVT, able to undermine the hemostatic balance along the vascular tree, pulling it towards an inflammatory and hypercoagulable condition.

D-dimer (DD) levels are the sum of the following three different trends: inflammation, coagulation activation and cancer mass activity [37,39]. This may explain why there is a very weak correlation between CEC (mainly related to inflammation and CRP) and DD in patients with hepatic disease, in which, differently from MPN patients, DD is mainly influenced by cancer mass activity and coagulation activation, but not by inflammation, as also shown by the low CRP level (see Table 1). This also contributes to explaining why there is an inverse correlation between CRP and thrombosis recanalization in the LC and HCC groups.

If we consider CEC as the expression of a prothrombotic endothelial injury, as a co-causal trigger of inflammation-induced SVT, or simply as a mirror of inflammation, we can understand the inverse relationship between CEC and thrombosis recanalization observed in all the analyzed groups.

In MPN, the spleen is not an innocent bystander or a helpless target of the disease. Rosti et al. provided evidence that some endothelial cells from the spleen and splenic veins of patients with MF bear the JAK2V617F mutation, probably contributing to sustaining and transforming MF [40]. As also shown in some experiments on mice, an MF spleen contains a lot of hematopoietic precursor cells and endothelial cells carrying the JAK2 mutation. Experimental data on mice document the existence of MF stem cells (MF-SCs) that reside in the spleen of animals with MF, and demonstrate that these MF-SCs retain a differentiation program identical to that of normal hematopoietic stem cells [41]. The spleen seems, therefore, to be a disease reservoir and a source of inflammatory mediators, sustaining the disease itself. This might explain why in MPN patients receiving RUXO or HU, the CEC levels decrease when SV reduction improves. In other words, CEC and SV decrease together, probably because they are different aspects of disease activity and burden. ANOVA testing found differences in the three treatment groups, only with reference to CEC reduction (*p* = 0.04) and SV reduction (*p* = 0.03), because RUX and HU also exhibit cytotoxic activity on MPN, reducing disease burden.

The SV reduction in MPN patients is not only related to splanchnic vein recanalization and portal pressure normalization, but also to disease burden and inflammation reduction, as shown in the current work; an inverse relationship between CRP and splenic volume reduction was observed in the RUX group and in the HU group, and a direct relationship between splenic volume reduction and the size of thrombosis recanalization was observed in the RUX group and in the HU group.

In patients with LC and HCC, in which the mechanism of SVT development is quite different, the relationship between CEC and SV reduction is absent. However, inflammation is also an important factor in determining SVT in these patients, as suggested by the inverse correlation between CRP and thrombosis recanalization (Figure 1, Figure 2 and Figure 3).

In our study, all the patients with SVT achieved higher thrombosis recanalization and a greater decrease in CRP and inflammation. In the patients with MPN, much higher thrombosis recanalization of SVT, and a much larger decrease in CRP (and inflammation) and disease burden was observed. RUX had a quicker and more powerful anti-inflammatory effect than hydroxyurea [39,42]. RUXO had higher cytotoxicity on the disease clone, leading to better recanalization of thrombosis than the HU treatment (*t*-test *p* = 0.01), with the highest percentage of thrombosis recanalization and SV reduction observed among the analyzed groups. This is corroborated by the literature; a metanalysis on PV and MF patients confirmed that ruxolitinib was efficient in reducing splenomegaly and lowering the rate of thrombosis [43]. In the phase II study (SVT-RUXO), in the subset of SVT-MPN patients, the researchers proved that RUXO not only reduced the spleen volume, but also stabilized the degree of esophageal varices, and reduced the extent of thrombosis and the resistivity indices of the hepatic and splenic intraparenchymal arteries [44].

We are aware of the main limitations of this study, which are as follows: the small sample size, the retrospectivity of the study, and the exclusive presence of MF in group A (treated with RUX). However, the collection of biological data in patients with thrombosis in an atypical site—moreover occurring as a manifestation of a low-incidence neoplasm—is, in our opinion, preparatory to more structured studies, both translational and clinical.

In conclusion, the splanchnic vein district is a very particular sector of the venous circle, characterized by continuous antigenic stimulation and inflammatory challenges. These stimuli make the splanchnic vein district prone to developing SVT in particular conditions, such as solid gastrointestinal neoplasms and MPN. Here, we have found an increase in CEC and SV in patients with SVT, mainly affected by MPN. These findings are probably not only related to thrombosis extension, vascular damage and portal hypertension, but they might reflect the disease course and treatment response. According to our findings, we can consider both increased spleen volume and high CEC levels as different aspects of MPN, able to perpetuate the disease itself. Monitoring CEC levels during cytoreductive and anticoagulant treatments may, thus, be useful to improve SVT outcome in patients with MPN.

## Figures and Tables

**Figure 1 jpm-12-00364-f001:**
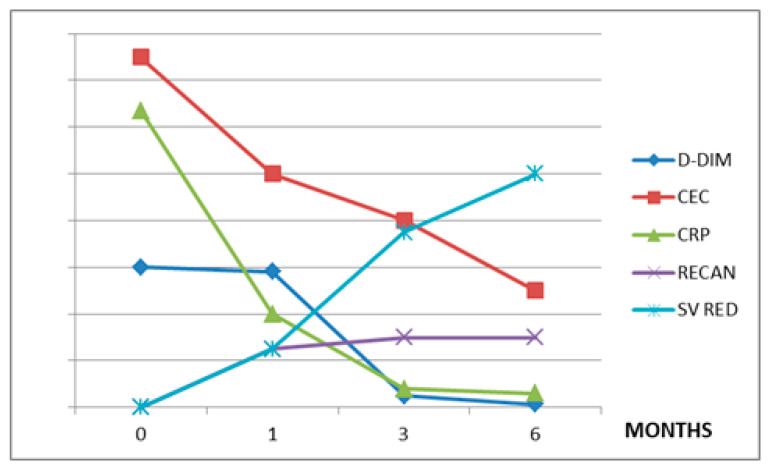
Ruxolitinib (A) group. CEC: circulating endothelial cells, CRP: C-reactive protein, D-DIM: d-dimers, RECAN: % of recanalization, SV: % of splenic volume reduction. Values on y-axis are in arbitrary measures.

**Figure 2 jpm-12-00364-f002:**
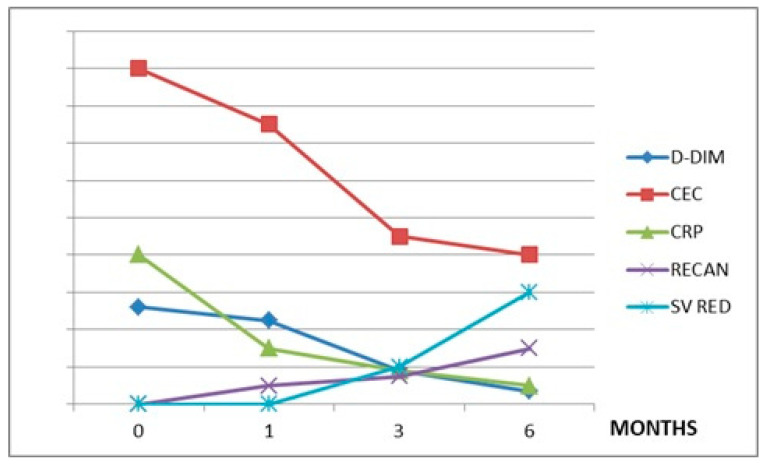
Hydroxyurea (B) group. CEC: circulating endothelial cells, CRP: C-reactive protein, D-DIM: d-dimers, RECAN: % of recanalization, SV: % of splenic volume reduction. Values on y-axis are in arbitrary measures.

**Figure 3 jpm-12-00364-f003:**
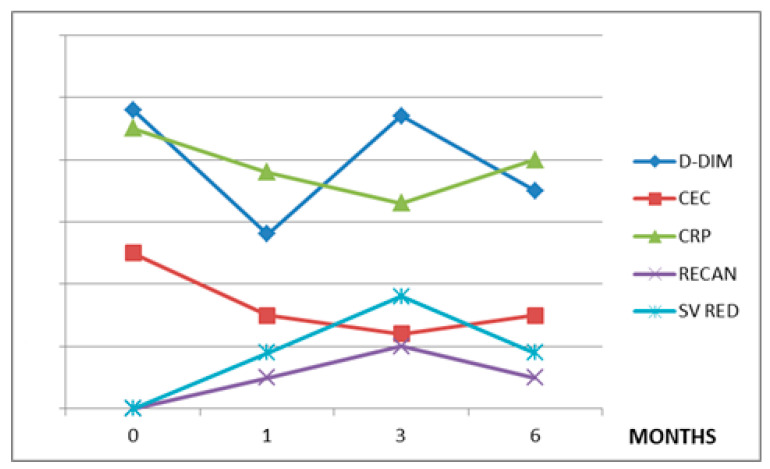
VKA (C) group. CEC: circulating endothelial cells, CRP: C-reactive protein, D-DIM: d-dimers, RECAN: % of recanalization, SV: % of splenic volume reduction. Values on y-axis are in arbitrary measures.

**Table 1 jpm-12-00364-t001:** Patient characteristics at SVT onset.

	A (RUXOLITINIB)	B (HU)	C (VKA)
PATIENTS, n	4	5	6
MEDIAN AGE, years (range)	45 (35–51)	44 (36–55)	60 (55–65)
SEX (M/F)	0/4	0/5	5/1
MF, n	4	1	0
PV, n	0	2	0
ET, n	0	2	0
LC, n	0	0	3
HCC, n	0	0	3
JAK2 V617F positive, n	4	5	0
THROMBOPHILIA, n	1	1	0
MEDIAN HB g/dL (range)	11.5 (11–12.5)	12.5 (10–14.5)	10 (9–11.5)
MEDIAN PLT/mmc (range)	90,000 (70,000–150,000)	150,000 (110,000–250,000)	40,000 (30,000–60,000)
MEDIAN WBC/mmc (range)	10,000 (4,–15,000)	9000 (4000–13,000)	6000 (3000–7500)
PVT, n	4	5	6
SVT, n	2	3	3
MVT, n	1	0	0
DD µg/mL (range)	6000 (5000–11,000)	5200 (4000–9000)	4800 (2000–5000)
CEC/µL (range)	1500 (1000–2500)	1800 (1000–2000)	2500 (1500–3000)
CRP mg/L (range)	127 (95–150)	80 (70–100)	45 (15–60)
SYS SYM (TOT PAT n)	4	4	6

CEC: circulating endothelial cells, CRP: C-reactive protein, D-D: d-dimers, ET: essential thrombocythemia, HCC: hepatocellular carcinoma, HB: hemoglobin, LC: liver cirrhosis, MF: myelofibrosis, MVT: mesenteric vein thrombosis, PLT: platelets, PV: polycythemia vera, PVT: portal vein thrombosis, SVT: splanchnic vein thrombosis, SYS SYM: systemic symptoms, WBC: white blood cells.

**Table 2 jpm-12-00364-t002:** Parameters observed during treatment (median values with standard deviations, SD).

GROUP	A (RUXO)				B (HU)				C (VKA)			
MONTH	0	1	3	6	0	1	3	6	0	1	3	6
DD µg/mL	6000	5800	500	130	5200	4500	1800	700	4800	2800	4700	3500
CEC/µL	1500 (SD 407.9)	1000	800	200 (SD 143.5)	1800 (SD 518.6)	1500	900	800 (SD 310.1)	2500	1500	1200	1500
CRP mg/L	127 (SD 47.6)	40	8	6 (SD 1.9)	80 (SD 47.9)	30	18	10 (SD 3.29)	45	38	33	40
RECAN %	0	62,5	75	75 (SD 8.9)	0	20	30	60 (SD 27.3)	0	9	18	9
SV RED %	0	25	75	100 (SD 17.0)	0	0	20	60 (SD 18.6)	0	0	0	0
SYS SYM (total n. pat. At each time)	0	4	4	4	0	0	1	3	0	0	0	0

**Table 3 jpm-12-00364-t003:** Pearson correlation among CEC and DD, CRP, thrombosis recanalization (RECAN) and splenic volume reduction (SV RED).

	A (RUXO)	B (HU)	C (VKA)
DD	0.8	0.98	0.3
CRP	0.7	0.9	0.9
RECAN	−0.7	−0.8	−0.9
SV RED	−0.96	−0.9	n.a.

RUXO: ruxolitinib, HU: hydroxyurea, VKA: vitamin k antagonists.

## Data Availability

The data presented in this study are available on request from the corresponding authors.

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
