# Peer review of "Circulating Endothelial Cell Levels Correlate with Treatment Outcomes of Splanchnic Vein Thrombosis in Patients with Chronic Myeloproliferative Neoplasms"

_jpm, 2022, doi:10.3390/jpm12030364_

Round 1

Reviewer 1 Report

Useful work has been done by the authors. The concept of CEC as a factor in thrombosis has been expanded. Unfortunately, there is no up-to-date analysis of the scientific literature.

Reviewer 2 Report

This is an interesting review on circulating Endothelial Cells (ECs, either viable, apoptotic or necrotic) as biomarkers of  splanchnic vein thrombosis, duly documented by the technical flow cytometry protocol based on identification with anti-CD 146 antibodies. These thrombotic events, very rare, are ferquently associated with mutations in ECs.

This report is concise, well-written and analyzed, although it concerns a highly specialized topic. In addition to variations of ECs over time and in response to therapies, the authors show the kinetics course of some biomarkers.

The limitations of the strudy, due to the small sample size for these rare clinical events, are correctly outlined.

I have no special comment on presentation, tables or figures which look appropriate for the message conveyed. The discussion section could be slightly condensed.

Concerning the specific comments, the abbreviations must be fully written at the first time they are cited in the text. This is not the case for MPN in the abstract (lines 23 and 25).

The standardized or consensus abbreviations should be used for biomarkers and units.

For example, the following abbreviations and units require attention: 

Line 164: µg better than mcg.

Table 1: µl better than MCL

D-D should be DD as in table 3

Table 2: DD rather than D-DIM (as in table 3) and µl better than mcl or µg in place of mcg

Reviewer 3 Report

In this study, Giordano et al. aims to establish a link between CEC and treatment outcomes in splanchnic vein thrombosis in patients with chronic myeloproliferative neoplasms.

It is an interesting study even if the number of patients is very low. However, some important data are missing to be really convince by the current study. The manuscript is hard to read, and the introduction should be rewriting. The introduction is longer than the result section, thus I suggest that the study should be presented as a letter and not as a full paper.

Major comments:

  • Introduction must be shortened. t is a way too long leading to the lost the main objective of the study in all the given information. A lot of percentage is given making the text a bit confusing. Some paragraphs must be reorganized to make it clearer.
  • The paragraph describing hydroxyurea (line 131-134) should be combined with the paragraph line 109-116 as the authors are describing a specific aspect of HU action mechanism (inhibition of P-selectin induction of tissue factor) without introducing what is HU. Moreover, references are missing regarding the role of HC on P-selectin and tissue factor. HU is given in several type of disorder like hemoglobinopathies. Are there any data on CEC and HU in those disorders?
  • Line 64-105: must be shortened as it is not adding any convincing data on the pertinence of performing the study.
  • Line 147: The authors mention that the study is a dicentric retrospective study but the centers are not mentioned.
  • Line 181: what is the accurate dosage of the drugs? What does die mean? Were the dosages adjusted regarding the patient’s weight?
  • Table 1: How do the authors explain the difference regarding the sex ratio difference in between the group? Could the difference influence the results? It is well known that men and women have different inflammatory profile.
  • It is not well stated if the patients were untreated by HU and RUX at inclusion day? If not, since when have they been treated?
  • HU is a well known cytostatic. It would have been relevant to follow up the blood cell number. The question of the compliance is also essential. For HU treatment, MCV is a good marker of compliance.
  • The main experimental tool used by the authors is flow cytometry. It is very disappointing that they are not showing any representative figure of flow cytometry.
  • Regarding CEC detection, how the authors can be sure that the CD146 are CEC and no other circulating cells expressing CD146 like B, T or NK cells? Why did they not used a specific marker of endothelial cells such as PECAM-1?
  • Line 251-255. The authors state that the high levels of CRP are due to inflammation state of their patients. They should have analyzed other markers of inflammation such as TNFa or IL1 that are known to activate the endothelium.
  • Did the authors perform statistical analysis to follow the parameters during the longitudinal study? In the tables or graph, no sign of statistical differences appears.

Minor comments:

  • The authors should be careful with the use of abbreviations. e.g.:

-Line 25: MPN must be defined in the abstract.

-Line 32: LC and HCC must be defined

-HU and RUX are defined too late

-circulating endothelial cells (CEC) such as CRP are written too many times

  • No reference for the used antibodies
  • Line 246: PCR should be changed for CRP
  • Line 271, 276, 291: errors in the text

Author Response

We are grateful to the Reviewers for their comments on our work. We changed our manuscript according to the reviewer’s comments. The sentences that have been changed were written with a red character in order to simplify the review of the paper.

Reviewer #3:

In this study, Giordano et al. aims to establish a link between CEC and treatment outcomes in splanchnic vein thrombosis in patients with chronic myeloproliferative neoplasms.

It is an interesting study even if the number of patients is very low. However, some important data are missing to be really convince by the current study. The manuscript is hard to read, and the introduction should be rewriting. The introduction is longer than the result section, thus I suggest that the study should be presented as a letter and not as a full paper.

We are grateful to the reviewer for appreciating our work.

We have now rewritten the introduction section reducing its length and making it easier to read.

Major comments:

  • Introduction must be shortened. t is a way too long leading to the lost the main objective of the study in all the given information. A lot of percentage is given making the text a bit confusing. Some paragraphs must be reorganized to make it clearer.

Thank you, we have now reduced the lenght of the introduction and rephrased some paragraphes.

  • The paragraph describing hydroxyurea (line 131-134) should be combined with the paragraph line 109-116 as the authors are describing a specific aspect of HU action mechanism (inhibition of P-selectin induction of tissue factor) without introducing what is HU. Moreover, references are missing regarding the role of HC on P-selectin and tissue factor. HU is given in several type of disorder like hemoglobinopathies. Are there any data on CEC and HU in those disorders?

Thank you, we modified the text accordingly and we added two citations about the role of HU on P-selectin and tissue factor (references 23 and 24).

  • Line 64-105: must be shortened as it is not adding any convincing data on the pertinence of performing the study.

Thanks for the advice, we modified and reduced the section of the text.

  • Line 147: The authors mention that the study is a dicentric retrospective study but the centers are not mentioned.

Thank for noticing; we specified in the text the two centers involved (Division of Hematology and Division of Gastroenterology in Campobasso, Molise region, Italy).

  • Line 181: what is the accurate dosage of the drugs? What does die mean? Were the dosages adjusted regarding the patient’s weight?

Thanks for the comment, we added the specification in the text. The dosage of the drugs was in accordance to the technical data sheet and was adjusted taking into account the hematological values and the patient’s weight.

  • Table 1: How do the authors explain the difference regarding the sex ratio difference in between the group? Could the difference influence the results? It is well known that men and women have different inflammatory profile.

Table 1 reports on patients’ characteristics, the observation that subjects with SVT secondary to MPN were only women, mainly with JAK-2 mutation is very interesting, we thank reviwer for this. In the discussion section, previously focused only on the role of JAK2-mutation, we have now added a paragraph on the gender related risk of thrombosis based on gender specific risk factors.

Some of the proposed mechanisms of how estrogens regulate inflammation include responses mediated via nuclear/membrane bound estrogen receptors, stimulation of toll like receptor ligand on dendritic cells and macrophages resulting in increased expression of pro-inflammatory cytokines, decreasing inhibitory PI3K signaling and Akt phosphorylation in macrophages, and effects on NF-κB transcriptional complexes.

  • It is not well stated if the patients were untreated by HU and RUX at inclusion day? If not, since when have they been treated?

The patients were included at the diagnosis of the splanchnic vein thrombosis, so at inclusion they were untreated; the therapy was started as soon as possible after diagnosis, within a few days.

  • HU is a well known cytostatic. It would have been relevant to follow up the blood cell number. The question of the compliance is also essential. For HU treatment, MCV is a good marker of compliance.

Thanks for the comment. Although not shown, we performed the blood count exams at every visit and collected the data. The values of the blood exams were consistent with the cytoreductive therapy and, in case of inadequate control of Hct or hematological toxicities, we adjusted the dose.

  • The main experimental tool used by the authors is flow cytometry. It is very disappointing that they are not showing any representative figure of flow cytometry.

We agree with the reviewer. We are aware that showing the image of flow cytometry would have enriched the work; unfortunately, for technical and logistical reasons, it was not possible to recover the images produced, even if the data of the events were saved. We tried to compensate, focusing on the descriptive quality of the data.

  • Regarding CEC detection, how the authors can be sure that the CD146 are CEC and no other circulating cells expressing CD146 like B, T or NK cells? Why did they not used a specific marker of endothelial cells such as PECAM-1?

In the materials section, we have described the way the experiment has been performed. For the detection of CEC ,we have followed a previously published protocol, involving these  specific markers: CD45,CD133 and CD146.

  • Line 251-255. The authors state that the high levels of CRP are due to inflammation state of their patients. They should have analyzed other markers of inflammation such as TNFa or IL1 that are known to activate the endothelium.

We agree with the reviewer, levels of TNF and IL1 should add important info to the current study, however we did not analyse these markers.

  • Did the authors perform statistical analysis to follow the parameters during the longitudinal study? In the tables or graph, no sign of statistical differences appears.

The analysis performed did not show significant differences.

Minor comments:

The authors should be careful with the use of abbreviations. e.g.:

  • Line 25: MPN must be defined in the abstract.
  • Line 32: LC and HCC must be defined
  • HU and RUX are defined too late
  • circulating endothelial cells (CEC) such as CRP are written too many times
  • No reference for the used antibodies
  • Line 246: PCR should be changed for CRP
  • Line 271, 276, 291: errors in the text

We thank the reviewer for the corrections suggested; we accordingly corrected them in the text.

Reviewer 4 Report

The paper entitled “Circulating endothelial cell levels correlate with treatment out-2 comes of splanchnic vein thrombosis in patients with chronic 3 myeloproliferative neoplasms” is well structured and presented. The main topics of the paper are well introduced and the results and discussion are well organized.

The methods section must be completed, namely about D Dimers measurement, according to recent publication “the manufacturer and kit used for D-dimer testing, whether DDU or FEU are reported, the assay cut-off value, and also the absolute measuring units (e.g. mg/L, ng/mL). Ideally, if relevant to the study population (e.g. venous embolism), the sensitivity, specificity, and negative and positive predictive values of D-dimer assays should also be quoted in addition to the normal reference ranges”

Line 195 – adjust the units of FITC “ sodium heparin, was labeled with 10 mcl of a panel of fluorescein isothiocyanate (FITC)”

In table 2, the parameters observed during treatment (median values) should also present the range (min. – maximum) to help to understand the variability of data.

In figure 1 and 2, the y axis should be arbitrary units.

Round 2

Reviewer 3 Report

I would like to thank the authors as they really improved the manuscript. It is easier to follow. They replied convincingly to most of my comments. It is a very challenging study regarding the low frequency of those disorders and primary results are quite convincing.

Nevertheless, I still do have major comments:

  • In the groups A and B: The authors state that the difference is not significative between D0 et 6 months regarding the difference parameters they are analyzing. However, CEC and CRP seem to decrease more than a half. On the contrary, RECAB and SV increase. What about standard deviation between patients? That might explain the absence of significance.
  • My main concern is regarding the flow cytometry data. I am willing to trust the result but without any figure, it is very difficult to appreciate the result and their analysis made by the authors. You cannot present western blot quantification without the picture related. It is the same for flow cytometry.

Author Response

  • In the groups A and B: The authors state that the difference is not significative between D0 et 6 months regarding the difference parameters they are analyzing. However, CEC and CRP seem to decrease more than a half. On the contrary, RECAB and SV increase. What about standard deviation between patients? That might explain the absence of significance.

Thanks for that question, it's definitely important to clarify that this isn't a standard deviation issue. We had chosen to omit the information just to aid readability of the table, but now it has been added over the 0- and 6-month values of groups A and B. We find it to be a good solution.

  • My main concern is regarding the flow cytometry data. I am willing to trust the result but without any figure, it is very difficult to appreciate the result and their analysis made by the authors. You cannot present western blot quantification without the picture related. It is the same for flow cytometry.

With the utmost honesty and transparency, in spite of numerous attempts made during the revision extension period, we are afraid to report that the flow cytometry figures are not recoverable due to an irreversible technical error. We've responded so far to major and minor comments from four reviewers, and that is our only element we're unable to improve.
Obviously, we take full responsibility for the data produced, which we believe to be reliable and worthy of publication. We understand at the same time the importance of supporting these data with an image, so we defer the omission to the judgment of the reviewer and the editor. 

Round 3

Reviewer 3 Report

I do want to thank the authors who did the best to provide changes that contribute to greatly improve the manuscript. I do still think the major piece of the puzzle is missing.